# Predictive Value of Temporal Muscle Thickness for Sarcopenia after Acute Stroke in Older Patients

**DOI:** 10.3390/nu14235048

**Published:** 2022-11-27

**Authors:** Ayano Nagano, Akio Shimizu, Keisuke Maeda, Junko Ueshima, Tatsuro Inoue, Kenta Murotani, Yuria Ishida, Naoharu Mori

**Affiliations:** 1Department of Nursing, Nishinomiya Kyoritsu Neurosurgical Hospital, 11-1 Imazuyamanaka-cho, Nishinomiya 663-8211, Japan; 2Department of Palliative and Supportive Medicine, Graduate School of Medicine, Aichi Medical University, 1-1 Yazakokarimata, Nagakute 480-1195, Japan; 3Department of Health Science, Faculty of Health and Human Development, The University of Nagano, 8-49-7 Miwa, Nagano 380-8525, Japan; 4Department of Geriatric Medicine, National Center for Geriatrics and Gerontology, 7-430 Morioka, Obu 474-8511, Japan; 5Department of Nutrition Service, NTT Medical Center Tokyo, 5-9-22 Higashi-Gotanda, Tokyo 141-8625, Japan; 6Department of Physical Therapy, Niigata University of Health and Welfare, 1398 Shimami-cho, Niigata 950-3198, Japan; 7Biostatistics Center, Kurume University, 67 Asahimachi, Kurume 830-0011, Japan; 8Department of Nutrition, Aichi Medical University Hospital, 1-1 Yazakokarimata, Nagakute 480-1195, Japan

**Keywords:** subacute stroke, sarcopenia, rehabilitation, disability

## Abstract

The assessment of sarcopenia is part of the nutritional assessment index and is essential in stroke management. This study aimed to identify and validate cutoff values of temporal muscle thickness (TMT) measured using computed tomography to identify sarcopenia after acute stroke. The participants were patients with stroke aged ≥65 years who were admitted to rehabilitation units. The recruited patients were randomly divided into the calculation and validation cohort. In the calculation cohort, TMT cutoff values for identifying sarcopenia were calculated using receiver operating characteristic analysis. The obtained values were validated in the validation cohort using sensitivity and specificity. The calculation cohort included 230 patients (125 men, mean age, 77.2 ± 7.2 years), whereas the validation cohort included 235 patients (125 men, mean age, 76.4 ± 6.95 years). The TMT cutoff values for identifying sarcopenia and low skeletal muscle index were the same: 3.83 mm for men and 2.78 mm for women. The TMT cutoff value for identifying sarcopenia showed a sensitivity and specificity of 0.642 and 0.750, respectively, for men, and 0.660 and 0.567, respectively, for women. We identified a valid cutoff value of temporal muscle thickness for identifying sarcopenia after acute stroke. TMT is easy to measure and may be useful for the early detection of sarcopenia.

## 1. Introduction

Stroke has a high global incidence. It can lead to various disabilities and a decreased ability to perform activities of daily living (ADL). Undernutrition and sarcopenia are negative factors for the recovery of physical function and ADL performance ability after acute stroke [1,2,3,4,5,6]. Stroke-related sarcopenia occurs due to immobility and decreased activity, as well as stroke-related muscle tissue changes such as inflammation, denervation, sympathetic activation, and shifts in muscle fiber type [7]. In addition, undernutrition and sarcopenia are associated, and the coexistence of undernutrition and sarcopenia is frequently observed in elderly patients [8]. In stroke, impaired consciousness, dysphagia, and functional impairment of the upper limbs can lead to poor nutritional intake [9,10], which can easily worsen nutritional status [3], and undernutrition can worsen sarcopenia [11]. An assessment of muscle mass is also important in the diagnostic criteria for malnutrition [12]. Therefore, it is essential to identify, prevent, and treat sarcopenia and undernutrition in patients with stroke. Sarcopenia is often diagnosed using criteria such as the Asian Working Group for Sarcopenia (AWGS) 2019 [13] and European Working Group on Sarcopenia in Older People 2 [14]. These criteria diagnose sarcopenia using indices of muscle strength, function, and muscle mass. In patients with stroke, it is often difficult to assess muscle strength and physical function due to paralysis and impaired consciousness. Dual-energy X-ray absorptiometry and bioelectrical impedance analysis (BIA), the gold standards for muscle mass assessment, are not yet widely used in clinical practice. Therefore, a sarcopenia assessment method that is easy to use clinically in patients with stroke is needed.

In recent years, temporal muscle thickness (TMT) has gained attention as a surrogate marker for identifying post-stroke function and prognosis [15]. TMT correlates with sarcopenia risk [16] and hand grip strength [17] in patients with stroke. A low TMT has been found to be associated with significantly decreased survival [18], and severe dysphagia [19] in patients with acute stroke. Katsuki et al. reported TMT cutoff values of 4.9 mm for women and 6.7 mm for men when identifying physical function at 3 months post-stroke in patients with subarachnoid hemorrhage under 75 years of age [20]. TMT has also been shown to be associated with nutritional status in older adults [21] and progression-free survival in patients with head and neck squamous cell carcinoma [18]. TMT can be measured using computed tomography (CT), magnetic resonance imaging (MRI), and ultrasonography. Since CT and MRI are routinely performed in patients with stroke, assessing sarcopenia based on TMT can enable early diagnosis and intervention and may contribute to improving the clinical outcomes of patients with acute stroke. However, no TMT cutoff values have been reported for identifying sarcopenia or skeletal muscle loss in patients after acute stroke. The purpose of this study was to calculate and validate the cutoff values of TMT for identifying sarcopenia in older Japanese patients with stroke.

## 2. Materials and Methods

### 2.1. Participants

This cross-sectional study was conducted at the Hamamatsu City Rehabilitation Hospital in Shizuoka, Japan. This hospital provides inpatient rehabilitation services and care covered by the Japanese national health insurance system [22]. The participants were patients aged ≥65 years who were admitted to rehabilitation units after acute stroke between June 2019 and June 2021. Patients were excluded if it was not possible to measure their body composition using BIA. This study was approved by the Ethics Committee of the Hamamatsu City Rehabilitation Hospital (ID:20-26). The requirement for informed consent was waved by the ethics committee because of the retrospective study design. Patients could withdraw from the study at any time using the opt-out feature on the study website.

### 2.2. Temporal Muscle Thickness

TMT was evaluated on CT images (window width: 100 mm and window level: 35 mm) obtained on the day of admission to the hospital using the method described by Nozoe et al. [16] TMT was measured at the level of the orbital roof, and the Sylvian fissure was used as a reference point to determine the anterior–posterior orientation. TMT was separately evaluated on the left and right sides in all patients by one registered dietician and one radiological technologist using Image J software (Ver. 1.52u) [23]. The TMT values of each side were summed and divided by two to obtain the mean TMT for each patient.

### 2.3. Sarcopenia Parameters

Sarcopenia was diagnosed according to the AWGS 2019 criteria [13]. Hand grip strength was measured using a Jamar dynamometer (MG-4800 digital grip strength meter; CHARDER Electronic, Taichung, Taiwan). The hand grip strength was measured separately for the left and right hands with the participant seated with their elbow at 90°, and the highest value from three measurements for each side was used for analysis. Muscle mass was calculated from the impedance measured using the bioimpedance analyzer (Inbody s10; InBody Japan, Tokyo, Japan). To minimize the influence of the measurement device, we used Yamada’s formula: $ALM=\left(0.6947\times \left(\frac{Ht2}{Z50}\right)\right)+\left(-55.24\times \left(\frac{Z250}{Z5}\right)\right)+\left(-10,940\times \left(\frac{1}{Z50}\right)\right)+51.33\;\mathrm{for}\;\mathrm{men}\;\mathrm{and}\;ALM=\left(0.6144\times \left(\frac{Ht2}{Z50}\right)\right)+\left(-31.61\times \left(\frac{Z250}{Z5}\right)\right)+\left(-9332\times \left(\frac{1}{Z50}\right)\right)+37.91\;\mathrm{for}\;\mathrm{women}$ [24]. The skeletal muscle index (SMI) was calculated as the ratio between the appendicular skeletal muscle mass and the height squared. Based on the cutoff mentioned in the AWGS 2019 criteria [13], women with an SMI < 5.7 kg/m^2^ and a hand grip strength < 18 kg and men with an SMI < 7.0 kg/m^2^ and a hand grip strength < 28 kg were diagnosed with sarcopenia. Gait speed and balance were not evaluated, as patients often have difficulties walking or standing after stroke. An SMI of <5.7 kg/m^2^ in men and <7.0 kg/m^2^ in women was considered a “low SMI” and a hand grip strength < 28 kg in men and <18 kg in women was considered a “low hand grip strength”.

### 2.4. Other Parameters

Stroke severity was assessed using the modified Rankin Scale [25]. The modified Rankin Scale score at stroke onset and time (in days) from stroke onset to rehabilitation hospital admission were determined from the records from the previous hospital. Nutritional status was assessed using the Mini Nutritional Assessment—Short Form, which is a tool for screening the nutritional status of older adults. Its scores range from 0 to 14, with scores of 11 or less indicating a poor nutritional status [26]. Comorbidities were assessed using the Charlson Comorbidity Index [27]. The ability to perform ADL was assessed by physical or occupational therapists using the Functional Independence Measure [28], which consists of motor (13 items) and cognitive (5 items) subscales. For each item, the patient’s independence level was graded on a scale from total assistance (1 point) to complete independence (7 points), and the total score ranged from 18 to 126 [28]. Each evaluation was performed within 3 days of admission. C-reactive protein values were obtained from blood samples taken at the time of admission.

### 2.5. Statistical Analysis

The recruited patients were randomly divided into two 1:1 cohorts: the calculation cohort, in which the cutoff value for identifying sarcopenia was calculated, and the validation cohort, in which the cutoff value obtained from analysis of the other cohort was validated. Descriptive statistics were used to describe patient characteristics in each cohort. Continuous and ordinal data are presented as mean ± standard deviation and median (25, 75 percentiles), respectively. Categorical data are expressed as frequencies and percentages. In the calculation cohort, TMT cutoff values for identifying low hand grip strength, low SMI, and sarcopenia were calculated using receiver operating characteristic analysis. In addition, the sensitivity, specificity, area under the curve (AUC), positive predictive value (PPV), and negative predictive value (NPV) for the obtained TMT cutoff values were calculated in the validation cohort.

## 3. Results

After excluding 20 patients whose muscle mass could not be measured using BIA, 465 patients were finally included. Of them, 230 (125 man, mean age, 77.2 ± 7.2 years) were included in the calculation cohort (Table 1). The mean TMT in this cohort was 3.98 ± 1.52 mm in men and 3.07 ± 1.20 mm in women. In the calculation cohort, the TMT cutoff values for identifying low SMI and sarcopenia were the same: 3.83 mm for men and 2.78 mm for women (Figure 1). The cutoff value for identifying low hand grip strength was 3.83 mm for men, which was the same as that for low SMI and sarcopenia, while it was 3.08 mm for women (Figure 1). 

The validation cohort included 235 patients (125 men, mean age, 76.4 ± 6.95 years) (Table 2). The mean TMT for validation cohort was 4.09 ± 1.53 mm in men and 3.10 ± 1.20 mm in women. There was no statistically significant difference in characteristics between the two groups. The sensitivity and specificity of the cutoff values for identifying low SMI identified in the calculation cohort were 0.774 and 0.583, respectively, for men, and 0.700 and 0.517, respectively, for women. The AUC, PPV, and NPV, were 0.735, 0.557, and 0.778, respectively, for men, and 0.704, 0.547, and 0.674, respectively, for women. The sensitivity and specificity of the cutoff values for identifying sarcopenia identified in the calculation cohort were 0.642 and 0.750, respectively, for men, and 0.660 and 0.567, respectively, for women. The AUC, PPV, and NPV, were 0.726, 0.654, and 0.740, respectively, for men, and 0.681, 0.559, and 0.667, respectively, for women (Table 3).

## 4. Discussion

In this study, we found a valid TMT cutoff value that predicts low SMI sarcopenia after acute stroke in older patients. We randomly divided older patients who had experienced acute stroke into two cohorts, calculated cutoff values for identifying low SMI and sarcopenia in one cohort, and validated the cutoff values in the other cohort. The TMT cutoff value for identifying low SMI and sarcopenia was 3.83 mm for men and 2.78 mm for women. The sensitivity, specificity, and AUC for identifying low SMI were 0.774, 0.583, and 0.735, respectively, for men, and 0.700, 0.517, and 0.704, respectively, for women. The TMT cutoff value for identifying sarcopenia showed a sensitivity, specificity, and AUC of 0.642, 0.750, and 0.726, respectively, for men, and 0.660, 0.567, and 0.681, respectively, for women.

The TMT cutoff values for identifying low SMI and sarcopenia identified in this study were found to be valid. Steindl et al. reported a correlation between TMT and hand grip strength measured using MRI in patients with neurological disease (Pearson correlation coefficient = 0.649; *p* < 0.001) [17]. Furthermore, Cho et al. found a weak correlation between TMT measured using MRI and appendicular skeletal mass (r = 0.379, *p* = 0.001) [29]. A correlation between pre-stroke sarcopenia risk, assessed using the strength, assistance in walking, rise from a chair, climb stairs, and falls questionnaire and TMT, has also been reported [16]. Thus, TMT may be useful in assessing sarcopenia as an indicator of muscle strength and mass. In this study, cutoff values of TMT for identifying sarcopenia were determined and validated in different cohorts. The sensitivity and specificity were 0.642 and 0.750, respectively, for men, and 0.660 and 0.567, respectively, for women. The AUC was 0.726 for men and 0.681 for women, with moderate accuracy, and we consider the cutoff values to be useful for screening for sarcopenia. In this study, TMT was calculated from CT images, but TMT can be measured using ultrasonography and MRI as well [21,30]. Ultrasonography has been used for nutritional status assessment, and the technology has been improved [31,32,33]. Ultrasonography also has potential uses for swallowing assessment [34]. Ultrasonography and CT are commonly used in clinical practice, and TMT measurement may be an alternative method of sarcopenia assessment in older patients with stroke. 

Sex differences were observed in TMT in the present study. Sex differences in muscle mass are generally observed, with men having more muscle mass than women, and men and women have different cut-off values for muscle mass [13]. Yesil Cinkir et al. measured TMT using MRI in patients with newly diagnosed glioblastoma multiforme (median (range) age: 56 (18–79) years) and found that the median TMT was 4.7 mm (range: 2.8–6.6 mm) in women and 5.4 mm (range: 2.9–8.1) in men [35]. In addition, Katsuki et al. reported that, in patients with subarachnoid hemorrhage aged 75 years or younger (mean [range] age: 60.6 (32–74) years), the TMT cutoff values for identifying physical function at 3 months were 4.9 mm for women and 6.7 mm for men [20]. In our study, the cutoff values of TMT for sarcopenia were 2.78 mm for women and 3.83 mm for men. The mean TMT and cutoff values for identifying sarcopenia were lower in our study than in a previous study [20,35]. The patients in this study were older than 65 years of age (mean age 77.2 ± 7.2 years), whereas the patients in the previous report were much younger than those in this study, with a mean age of 56 or 60 years. TMT, and the thicknesses of other muscles, should be considered during the evaluation of age-related loss of muscle mass and sex differences in muscle mass.

The strength of this study is that, for the first time, we recognized TMT cutoff values for identifying low SMI and sarcopenia after acute stroke in older Japanese patients. In this study, cutoff values were calculated and validated using two different cohorts. Appropriate random sampling and reliable results were obtained. Sarcopenia affects the prognosis of patients with stroke [4,36,37]. Although it is difficult to measure motor and physical function in patients with stroke due to factors such as paralysis, head CT findings are easily obtainable in clinical practice, and the ability to use TMT for identifying sarcopenia may permit rapid intervention. Rehabilitation nutrition or nutritional management according to rehabilitation and rehabilitative practices that take nutritional status into account is necessary [38,39]. Although the accuracy and specificity of the TMT cutoff values identifying low SMI and sarcopenia shown in this study were not very high, the ability to assess the risk of sarcopenia from CT images, which are routinely obtained in patients with stroke, may be useful in clinical practice. Nevertheless, this study was conducted on Japanese patients, and the cutoff values of TMT may differ depending on race; therefore, further validation in non-Asian patients is needed. Although muscle mass is affected by age, age-adjusted cut-off values could not be calculated in this study due to the insufficient sample size. In addition, since sarcopenia is affected by various factors, such as hormonal changes and inflammation, it is necessary to consider the influence of these factors in the correlation between TMT and sarcopenia. However, the present study was conducted as a retrospective study with a limited sample size and information. Therefore, further validation is needed. Moreover, the validity of the cutoff values for identifying prognosis was not verified in our study. In a previous study, TMT was a predictor of severe dysphagia in patients with stroke [40], but an independent association with functional prognosis was not reported [16]. Future validation of the association of TMT with prognosis in patients with stroke is warranted.

## 5. Conclusions

In this study, we calculated and validated TMT cutoff values for identifying sarcopenia in older Japanese patients who had experienced acute stroke. TMT cutoff value for identifying sarcopenia showed a sensitivity, specificity, and AUC of 0.642, 0.750, and 0.726, respectively, for men, and 0.660, 0.567, and 0.681, respectively, for women. Assessing sarcopenia using TMT measured using CT may be clinically useful.

## Figures and Tables

**Figure 1 nutrients-14-05048-f001:**
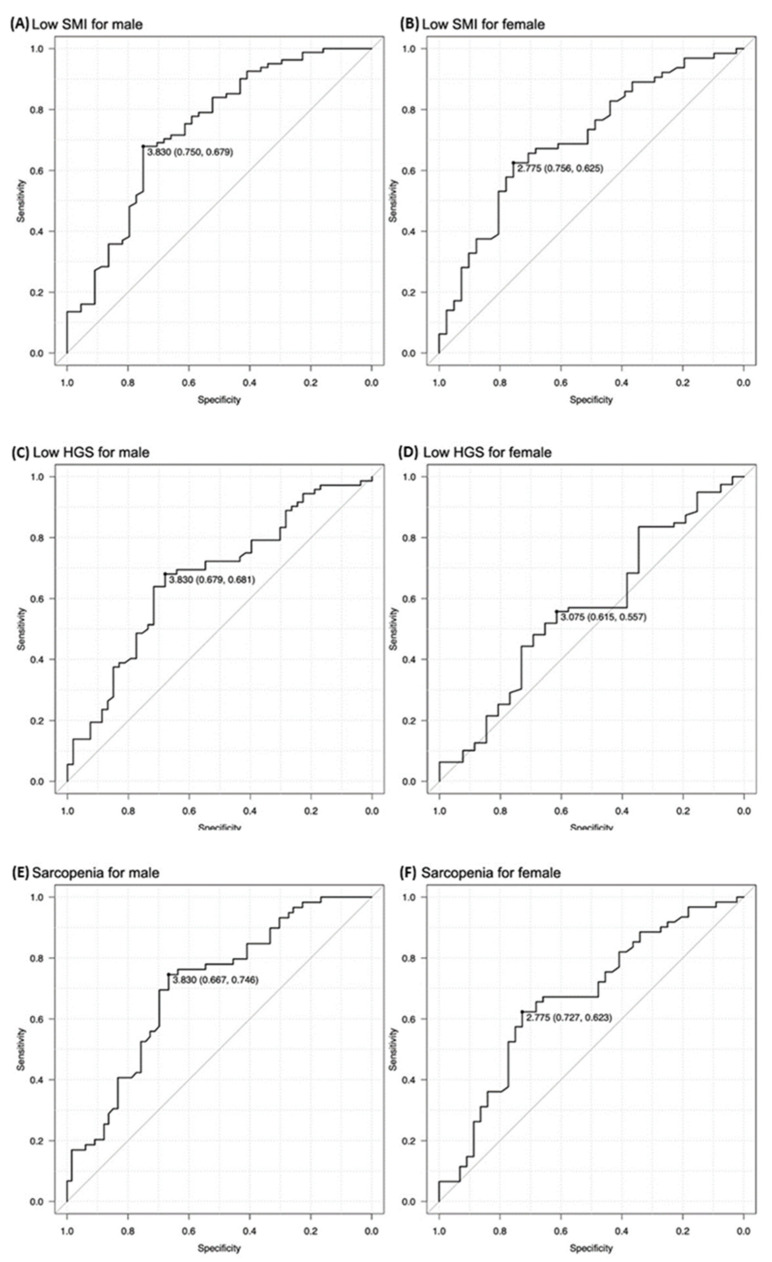
The temporal muscle thickness (TMT) cutoff values for identifying low skeletal muscle index (SMI), low hand grip strength (HGS), and sarcopenia. The TMT cutoff values for identifying low SMI were 3.83 mm for men and 2.78 mm for women (**A**,**B**, respectively). The TMT cutoff values for identifying low HGS were 3.83 mm for men and 3.08 mm for women (**C**,**D**, respectively). The TMT cutoff values for identifying sarcopenia were 3.83 mm for men and 2.78 mm for women (**E**,**F**, respectively).

**Table 1 nutrients-14-05048-t001:** Characteristics of the calculation cohort.

	Total Cohort n = 230
Sex, Male, n (%)	125 (54.3)
Age, year	77.2 ± 7.2
Type of stroke, n (%)	
- Cerebral infarction	136 (9.1)
- Cerebral hemorrhage	87 (37.8)
- Subarachnoid hemorrhage	7 (3.0)
Stroke onset to admission, days	26 (19–36)
Charlson Comorbidity Index score	1 (1–2)
Body mass index, kg/m^2^ C-reactive protein, mg/dL	21.5 ± 3.5 0.69 ± 1.60
Mini Nutritional Assessment—Short Form score	7 (5–9)
Modified Rankin Scale score at stroke onset	4 (3–4)
Functional Independence Measure score	72 (49–92)
SMI, kg/m^2^	6.12 ± 1.09
- Men	6.61 ± 1.08
- Women	5.54 ± 0.76
TMT, mm	3.57 ± 1.45
- Men	3.98 ± 1.52
- Women	3.07 ± 1.20
HGS, kg	20.2 ± 9.8
- Men	25.1 ± 9.1
- Women	14.3 ± 7.0

**Table 2 nutrients-14-05048-t002:** Characteristics of the validation cohort.

	Total Cohort n = 235
Sex, Male, n (%)	125 (53.2)
Age, year	76.4 ± 6.95
Type of stroke, n (%)	
-Cerebral infarction	144 (61.3)
-Cerebral hemorrhage	84 (35.7)
-Subarachnoid hemorrhage	7 (3.0)
Stroke onset to admission, days	26 (19–34)
Charlson Comorbidity Index score	1 (1–2)
Body mass index, kg/m^2^ C-reactive protein, mg/dL	21.6 ± 3.3 0.68 ± 1.90
Mini Nutritional Assessment—Short Form score	7 (5–9)
Modified Rankin Scale score at stroke onset	4 (3–4)
Functional Independence Measure score	77 (51–95)
HGS, kg -Men -Women	20.9 ± 9.9 26.3 ± 8.65 14.7 ± 7.3
SMI, kg/m^2^ -Men -Women	6.25 ± 1.04 6.83 ± 0.94 5.60 ± 0.70
TMT, mm -Men -Women	3.63 ± 1.48 4.09 ± 1.53 3.10 ± 1.20
Low SMI, n	135 (57.4)
Low TMT, n	103 (43.8)

**Table 3 nutrients-14-05048-t003:** Accuracy of low TMT for identifying low SMI and sarcopenia.

	Sensitivity	Specificity	AUC	PPV	NPV
Low SMI					
-Men	0.774	0.583	0.735	0.577	0.778
-Women	0.700	0.517	0.704	0.547	0.674
Sarcopenia					
-Men	0.642	0.750	0.726	0.654	0.740
-Women	0.660	0.567	0.681	0.559	0.667

Abbreviations: AUC: area under the curve, PPV: positive predictive value, NPV: negative predictive value.

## Data Availability

The data are not publicly available due to privacy.

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
