# Peer review of "Predictive Value of Temporal Muscle Thickness for Sarcopenia after Acute Stroke in Older Patients"

_nutrients, 2022, doi:10.3390/nu14235048_

Round 1

Reviewer 1 Report

The manuscript entitled "Predictive value of temporal muscle thickness for sarcopenia  after acute stroke in older patients" focused on the new diagnostic method of sarcopenia for patients with acute stroke. The study is interesting, however, the biggest problem is that what the relationship of this study with nutrient. It doesn't fit the aim and scope of this journal, and no correlate with all subject areas.

Author Response

We appreciate for the reviewer’s time and comments. The manuscript has been revised, and the changes were highlighted yellow in the revised version. Please find below our responses to your comments:

Comment #1: The manuscript entitled "Predictive value of temporal muscle thickness for sarcopenia after acute stroke in older patients" focused on the new diagnostic method of sarcopenia for patients with acute stroke. The study is interesting, however, the biggest problem is that what the relationship of this study with nutrient. It doesn't fit the aim and scope of this journal, and no correlate with all subject areas.

Response: Thank you for your comment. Sarcopenia is closely related to undernutrition, and their coexistence is frequently observed in the elderly. In recent years, the assessment of muscle mass has become an important part of the diagnostic criteria for malnutrition, and the diagnosis of sarcopenia is recognized as a necessary component of nutritional assessment. Nutrients is soliciting studies on novel methods for assessing malnutrition. We believe that our study is in line with the aim and scope of Nutrients. We believe "Malnutrition" as the appropriate subject.

We added additional background sentences regarding the association between malnutrition and sarcopenia in stroke in the Introduction section (page2 line47).

Stroke has a high global incidence. It can lead to various disabilities and a decreased ability to perform activities of daily living (ADL). Undernutrition and sarcopenia are negative factors for the recovery of physical function and ADL performance ability after acute stroke [1-6]. Stroke-related sarcopenia occurs due to immobility and decreased activity as well as stroke-related muscle tissue changes such as inflammation, denervation, sympathetic activation, and shifts in muscle fiber type [7]. In addition, undernutrition and sarcopenia are associated, and the coexistence of undernutrition and sarcopenia is frequently observed in elderly patients [8]. In stroke, impaired consciousness, dysphagia, and functional impairment of the upper limbs can lead to poor nutritional intake [9,10], which can easily worsen nutritional status [3] , and undernutrition can worsen sarcopenia[11]. Assessment of muscle mass is also important in diagnostic criteria for malnutrition [12]. Therefore, it is essential to identify, prevent, and treat sarcopenia and undernutrition in patients with stroke.

Reviewer 2 Report

The study by Nagano et al. recruited patients with stroke aged ≥65 years to identify and validate cutoff values of temporal muscle thickness after acute stroke-induced sarcopenia. In my opinion, the manuscript is well-written and helpful for the early detection of sarcopenia. I have only two comments: a) is there a relationship between sarcopenia development and nutrition in these patients? Can the author comment on this? 

b) Can the author normalize the clinical data by the age of each patient?

Author Response

Reviewer 2

We appreciate the reviewer for the insightful comments, which have helped us to improve our manuscript. The manuscript has been revised in accordance with the reviewer’s suggestions, and the changes were highlighted yellow in the revised version. Please find below our responses to your comments:

Comment #1: is there a relationship between sarcopenia development and nutrition in these patients? Can the author comment on this?

Response: Thank you for your valuable comments. As you pointed out, there is an association between sarcopenia development and nutrition in stroke patients. I added the new information on the association between sarcopenia and undernutrition in stroke patients in the Introduction (page2, line47) and Discussion sections (page7, line235).

Introduction

Stroke has a high global incidence. It can lead to various disabilities and a decreased ability to perform activities of daily living (ADL). Undernutrition and sarcopenia are negative factors for the recovery of physical function and ADL performance ability after acute stroke [1-6]. Stroke-related sarcopenia occurs due to immobility and decreased activity as well as stroke-related muscle tissue changes such as inflammation, denervation, sympathetic activation, and shifts in muscle fiber type [7]. In addition, undernutrition and sarcopenia are associated, and the coexistence of undernutrition and sarcopenia is frequently observed in elderly patients [8]. In stroke, impaired consciousness, dysphagia, and functional impairment of the upper limbs can lead to poor nutritional intake [9,10], which can easily worsen nutritional status [3], and undernutrition can worsen sarcopenia [11]. Assessment of muscle mass is also important in diagnostic criteria for malnutrition [12]. Therefore, it is essential to identify, prevent, and treat sarcopenia and undernutrition in patients with stroke.

Discussion

Although it is difficult to measure motor and physical function in patients with stroke due to factors such as paralysis, head CT findings are easily obtainable in clinical practice, and the ability to use TMT for identifying sarcopenia may permit rapid intervention; Rehabilitation Nutrition, that is, nutritional management according to rehabilitation and rehabilitative practices that take nutritional status into account, is necessary [38, 39].

Comment #2: Can the author normalize the clinical data by the age of each patient?

Response: Thank you for your comment. Performing multivariate analysis, age was an explanatory factor for TMT. However, due to insufficient sample size, age adjusted cutoff values could not be calculated. Therefore, we have added the the Limitation (page7 line242).

Although muscle mass is affected by age, age-adjusted cut-off values could not be calculated in this study due to insufficient sample size. Future studies are needed.

Reviewer 3 Report

The paper “Predictive value of temporal muscle thickness for sarcopenia after acute stroke in older patients” is a cross-sectional study investigating the existence of a positive correlation between temporal muscle thickness as measured by computed tomography and sarcopenia after acute stroke. The topic is of interest but some major revisions should be addressed before publication.

ABSTRACT

This section could be completed with an opening sentence that provides a brief background on the topic and highlights the importance and necessity of the present study.

INTRODUCTION

The background provided by the authors could be substantially improved. Sarcopenia is known to be associated with chronic systemic inflammation and de-regulation of the expression of several myokines. Can acute stroke affect this?

MATERIALS AND METHODS

2.1 Participants.

Sarcopenia is a multifactorial syndrome so many factors can determine its cause, severity, and progression. Therefore, it is important to specify which inclusion and exclusion criteria were adopted.

2.2 Statistical analysis.

No statistical tests were performed to check for the presence or absence of significance between the two cohorts of patients?

RESULTS

Insufficient data are provided to support the existence of a correlation between temporalis muscle thickness and sarcopenia. Were the patients not subjected to blood sampling? What were the values of vitamin D, PTH, and markers of inflammation? This section needs to be further investigated.

DISCUSSION

Implement this section by discussing any new findings provided.

Author Response

RESPONSE TO REVIEWERS

Reviewer 3

We appreciate the reviewer for his/her insightful comments, which have helped us to improve our manuscript. The manuscript has been revised in accordance with the reviewer’s suggestions, and these revisions have been highlighted.

Comment #1: ABSTRACT This section could be completed with an opening sentence that provides a brief background on the topic and highlights the importance and necessity of the present study.

Response: Thank you for your comment. I have added the text as suggested in the abstract.

Abstract: The assessment of sarcopenia is part of the nutritional assessment index and is essential to stroke management. This study aimed to identify and validate cutoff values of temporal muscle thickness (TMT) measured using computed tomography for identifying sarcopenia after acute stroke.

Comment #2: INTRODUCTION The background provided by the authors could be substantially improved. Sarcopenia is known to be associated with chronic systemic inflammation and de-regulation of the expression of several myokines. Can acute stroke affect this?

Response: Thank you for your valuable comments. As you pointed out, chronic systemic inflammation and catabolic activation due to excessive sympathetic activation are thought to affect stroke-related sarcopenia. We have added the additional information to the introduction section. (page2, line47)

Stroke has a high global incidence. It can lead to various disabilities and a decreased ability to perform activities of daily living (ADL). Undernutrition and sarcopenia are negative factors for the recovery of physical function and ADL performance ability after acute stroke [1-6]. Stroke-related sarcopenia occurs due to immobility and decreased activity as well as stroke-related muscle tissue changes such as inflammation, denervation, sympathetic activation, and shifts in muscle fiber type [7]. In addition, undernutrition and sarcopenia are associated, and the coexistence of undernutrition and sarcopenia is frequently observed in elderly patients [8]. In stroke, impaired consciousness, dysphagia, and functional impairment of the upper limbs can lead to poor nutritional intake [9,10], which can easily worsen nutritional status [3] , and undernutrition can worsen sarcopenia[11]. Assessment of muscle mass is also important in diagnostic criteria for malnutrition [12]. Therefore, it is essential to identify, prevent, and treat sarcopenia and undernutrition in patients with stroke.

Comment #3: MATERIALS AND METHODS 2.1 Participants. Sarcopenia is a multifactorial syndrome so many factors can determine its cause, severity, and progression. Therefore, it is important to specify which inclusion and exclusion criteria were adopted.

Response: I agree with your comments. However, we aimed to identify the TMT cutoff value that identifies sarcopenia in post-acute stroke, and because of the insufficient sample size, we did not perform a factorial analysis that affect sarcopenia. Therefore, all post-acute stroke patients for whom body composition could be measured by BIA were included.

Comment #4: 2.2 Statistical analysis. No statistical tests were performed to check for the presence or absence of significance between the two cohorts of patients?

Response: Thank you for your comment. In this study, we did not verify whether there was a significant difference between the two groups, because of our assumption of no significant difference in background between the two groups due to random sampling. In response to your suggestion, we compared the two groups using a t-test. As shown in the table below, there was no significant difference between the two groups. The fact that there was no significant difference between the two groups was added to the results (page6 line167).

The validation cohort included 235 patients (125 men, mean age, 76.4±6.95 years) (Table 2). The mean TMT for validation cohort was 4.09±1.53 mm in men and 3.10±1.20 mm in women. There was no statistically significant difference in characteristics between the two groups.

Calculation group

Validation group

p-value

Sex, male, n (%)

125 (54.3)

125 (53.2)

0.803

Age, year

77.2±7.2

76.4 ± 6.95

0.277

Type of stroke, n (%)

0.607

-Cerebral infarction

136 (59.1)

144 (61.3)

-Cerebral hemorrhage

87 (37.8)

84 (35.7)

-Subarachnoid hemorrhage

7 (3.0)

7 (3.0)

Stroke onset to admission, days

26 [19–36]

26 [19–34]

Charlson Comorbidity Index score

1 [1–2]

1 [1–2]

0.886

Body mass index, kg/m2

C-reactive protein, mg/dl

21.5±3.5

0.69±1.60

21.6 ± 3.3

0.68±1.90

0.665

0.913

Mini Nutritional Assessment – Short Form score

7 [5–9]

7 [5–9]

0.091

Modified Rankin Scale score at stroke onset

4 [4–4]

4 [3–4]

0.247

Functional Independence Measure score

72 [49–92]

77 [51–95]

0.301

SMI, kg/m2

6.12±1.09

6.25 ± 1.04

0.178

-Men

6.61±1.08

6.83 ± 0.94

0.880

-Women

5.54±0.76

5.60 ± 0.70

0.548

TMT, mm

3.57±1.45

3.63 ± 1.48

0.643

-Men

3.98±1.52

4.09 ± 1.53

0.552

-Women

3.07±1.20

3.10 ± 1.20

0.874

HGS, kg

20.2±9.8

20.9 ± 9.9

0.416

-Men

25.1±9.1

26.3 ± 8.65

0.256

-Women

14.3±7.0

14.7 ± 7.3

0.678

Comment #5: RESULTS Insufficient data are provided to support the existence of a correlation between temporalis muscle thickness and sarcopenia. Were the patients not subjected to blood sampling? What were the values of vitamin D, PTH, and markers of inflammation? This section needs to be further investigated.

Response: Thank you for your valuable comments. Unfortunately, this is a retrospective study, and vitamin D and PTH were not measured in the usual care of rehabilitation wards. However, since it is important to consider the effects of hormonal changes and other factors while discussing the relationship between TMT and sarcopenia, we added a sentence to the discussion (pg#, line#). As for the blood data, CRP was obtained, so it was added to Tables 1 and 2, and relevant sentences were added in Methods (page3 line133), Results (table1 and 2) and Discussion sections (page8 line 244).

Methods

C-reactive protein values were obtained from blood samples taken at the time of admission.

Results

Table 1. Characteristics of the calculation cohort (n=230)

Sex, male, n (%)

125 (54.3)

Age, year

77.2±7.2

Type of stroke, n (%)

-Cerebral infarction

136 (59.1)

-Cerebral hemorrhage

87 (37.8)

-Subarachnoid hemorrhage

7 (3.0)

Stroke onset to admission, days

26 [19–36]

Charlson Comorbidity Index score

1 [1–2]

Body mass index, kg/m2

C-reactive protein, mg/dl

21.5±3.5

0.69±1.60

Mini Nutritional Assessment – Short Form score

7 [5–9]

Modified Rankin Scale score at stroke onset

4 [4–4]

Functional Independence Measure score

72 [49–92]

SMI, kg/m2

6.12±1.09

-Men

6.61±1.08

-Women

5.54±0.76

TMT, mm

3.57±1.45

-Men

3.98±1.52

-Women

3.07±1.20

HGS, kg

20.2±9.8

-Men

25.1±9.1

-Women

14.3±7.0

Table 2. Characteristics of the validation cohort (n=235)

Sex, male, n (%)

125 (53.2)

Age, year

76.4 ± 6.95

Type of stroke, n (%)

-Cerebral infarction

144 (61.3)

-Cerebral hemorrhage

84 (35.7)

-Subarachnoid hemorrhage

7 (3.0)

Stroke onset to admission, days

26 [19–34]

Charlson Comorbidity Index score

1 [1–2]

Body mass index, kg/m2

C-reactive protein, mg/dl

21.6 ± 3.3

0.68 ± 1.90

Mini Nutritional Assessment – Short Form score

7 [5–9]

Modified Rankin Scale score at stroke onset

4 [3–4]

Functional Independence Measure score

77 [51–95]

HGS, kg

-Men

-Women

20.9 ± 9.9

26.3 ± 8.65

14.7 ± 7.3

SMI, kg/m2

-Men

-Women

6.25 ± 1.04

6.83 ± 0.94

5.60 ± 0.70

TMT, mm

-Men

-Women

3.63 ± 1.48

4.09 ± 1.53

3.10 ± 1.20

Discussion

Since sarcopenia is affected by various factors such as hormonal changes and inflammation, it is necessary to consider the influence of these factors in the correlation between TMT and sarcopenia. However, the present study was conducted as a retrospective study with limited sample size and information. Therefore, further validation is needed.

Comment #6: DISCUSSION Implement this section by discussing any new findings provided.

Response: Thank you for your comment. We have added the following sentences in Discussion. (page7 line 299)

The strength of this study is that for the first time we recognized TMT cutoff values for identifying low SMI and sarcopenia after acute stroke in Japanese older patients. In this study, cutoff values were calculated and validated using two different cohorts. Appropriate random sampling and reliable results were obtained.

Round 2

Reviewer 3 Report

The authors met my demands. The work can therefore be accepted for publication.